# Targeted Metagenomic Databases Provide Improved Analysis of Microbiota Samples

**DOI:** 10.3390/microorganisms12010135

**Published:** 2024-01-10

**Authors:** Agnes Baud, Sean P. Kennedy

**Affiliations:** Institut Pasteur, Université Paris Cité, Département de Biologie Computationnelle, F-75015 Paris, France

**Keywords:** microbiota, taxonomic binning, microbial diversity, databases, metagenomic analysis

## Abstract

We report on Moonbase, an innovative pipeline that builds upon the established tools of MetaPhlAn and Kraken2, enhancing their capabilities for more precise taxonomic detection and quantification in diverse microbial communities. Moonbase enhances the performance of Kraken2 mapping by providing an efficient method for constructing project-specific databases. Moonbase was evaluated using synthetic metagenomic samples and compared against MetaPhlAn3 and generalized Kraken2 databases. *Moonbase* significantly improved species precision and quantification, outperforming marker genes and generalized databases. Construction of a phylogenetic tree from 16S genome data in *Moonbase* allowed for the incorporation of UniFrac-type phylogenetic information into diversity calculations of samples. We demonstrated that the resulting analysis increased statistical power in distinguishing microbial communities. This study highlights the continual evolution of metagenomic tools with the goal of improving metagenomic analysis and highlighting the potential of the *Moonbase* pipeline.

## 1. Introduction

The human microbiota is comprised of trillions of microorganisms inhabiting the various niches in and on the human body and plays a fundamental role in health and disease [1]. The advent of high-throughput sequencing technologies and advanced bioinformatics tools has made possible comprehensive studies investigating the genomic diversity and functional potential of the human microbiota, alone with its interactions with the human host. One of the earliest observations of such studies has been the role of microbial diversity and its association with health [2]. As a whole, microbiota diversity increases in the years following birth. In the human intestine, this diversity is recognized as closely linked to proper immune system function, providing a barrier to pathogens and contributing to proper nutrient absorption [3]. In contrast, low diversity is considered to be the optimal composition of the vaginal microbiota. Here, the single genus of *Lactobacillus* serves in its protector role primarily by maintaining a low pH that is inhospitable to other organisms, including potential pathogens [4].

The introduction of high-throughput sequencing technologies revolutionized the study of human microbiota, moving beyond traditional culture-based techniques to describe the full diversity of microorganisms in the microbiota [5]. Early surveys of digestive tract microorganisms revealed that a large proportion of the organisms present, especially strict anaerobes, are poorly represented in databases owing to difficulties in isolation and culturing. Large groundbreaking studies, including MetaHIT and HMP, were initiated in order to gain a better understanding of the diversity of the human microbiota, as well as to generate a comprehensive catalog of organisms and genes from this environment [6,7]. A multitude of data from subsequent studies have been reported, leading to more comprehensive coverage in public databases. The development of advanced culturing techniques has also improved the isolation and sequencing of many species from the human gut [8]. 

Lower costs and improved technology have increased our ability to generate vast amounts of sequence data in a single experiment. These advances have facilitated large-scale comparative studies, allowing researchers to explore variations in microbiota across individuals, populations, and diseases. Coupled with increased sequencing data, bioinformatic tools have revolutionized the analysis of microbiota data by enabling the processing, classification, and interpretation of vast amounts of sequencing information. These tools play a pivotal role in translating raw sequencing data into meaningful biological insights, allowing researchers to discern microbial taxa, infer phylogenetic relationships, and estimate species abundances. However, the results generated by these tools are ultimately dependent on their robustness and accuracy. 

The correct identification of short sequencing reads in metagenomic samples, particularly from clinical studies, hinges on their comparison against sequences in public databases. The growth of these databases has enabled the identification of a greater proportion of reads in metagenomic studies but also introduced challenges. As database sizes increase, there are fewer unique markers for precise species and strain identification. Key community players or potential pathogens, often the most studied, are overrepresented in these databases, leading to biases in analysis, including abundance estimations. This overrepresentation is evident even in tools like MetaPhlAn (v3.0), which is designed for analyzing human microbiota samples. The accuracy of metagenomic analysis, whether using public or private databases, depends heavily on their quality and composition. Studies on human microbiota, including our group’s research on reproductive-aged women, demonstrate that tools analyzing low-biomass samples often yield results that do not accurately reflect biological function or experimental findings. Notable instances include reports of placental microbiota, which were later identified as artifacts from contamination or database biases, especially in low-bacterial-biomass environments [9,10,11].

A further challenge in microbiota research lies in accurately assessing the diversity and quantifying the abundance of microbial species within these communities. Enumeration of the number of unique taxa in a given sample is called α-diversity. Many of the current tools used to assess the α-diversity of the human microbiota have been adapted from ecology studies, including Shannon and Simpson indexes [12,13]. Another important aspect is how individuals or populations change, often in the context of some change, such as pregnancy, or treatment, including antibiotics. Such changes are called β-diversity and are commonly measured by translating observed differences in microorganism taxa into mathematical matrices of distances between samples. A commonly used method to calculate such distances for microbiota samples is the Bray–Curtis dissimilarity index [14].

A primary weakness of ecology-derived diversity measurements is that they do a poor job of integrating the functional diversity of a given bacteriological population. While both Shannon and Simpson indexes account for the total number of taxonomic units and the evenness of their distribution, neither considers genetic diversity nor phylogenetic distances. In two samples, each with 10 species, there would be no difference between one containing representatives of 10 different phyla and another containing all representatives from a single phylum. Faith’s phylogenetic diversity (PD) addresses this problem by considering the phylogenetic distances between members of a population in its calculation [15]. A similar problem is found in β-diversity with Bray–Curtis dissimilarity. Here, another solution linked to phylogenetic distance was proposed by Lozupone and Knight, Unifrac, which uses the distances on a phylogenetic tree generated from 16s rDNA sequences [16].

Here, we describe a hybrid use of MetaPhlAn3 (v3.0) [17] and Kraken2 (v.2.1.3) [18] to improve accuracy of bacterial species detection and quantification in complex communities. Taxonomic diversity was found to be particularly important in studying the perinatal vaginal microbiota, and we also show application of Weighted UniFrac [16] to shotgun metagenomic datasets.

## 2. Materials and Methods

### 2.1. Programing Environment

The workflow of Moonbase was coded using Snakemake (v. 7.16.1). Code, documentation, and required files are available from the GitLab Repository (https://gitlab.pasteur.fr/metagenomics/moonbase). Figures were generated using the plotly Python library. Analyses were performed in the Jupyter notebook environment [19]. External packages MetaPhlAn (v3.0) [17], Kraken (v2.1.3) [18], and Python3 (v3.8), along with their dependencies, were required. Moonbase required downloading additional files from the repository: A Kraken2 taxonomy directory, a multi-FASTA file, as well as a correspondence file between the headers from the multi-FASTA file and their associated taxID.

The ‘generalized microbial database’ used for comparing Kraken2 performance is the kraken2-microbial-fatfree kraken2 database (available at https://lomanlab.github.io/mockcommunity/mc_databases.html (accessed on 24 November 2020)) prepared from the archaea, bacteria, fungi, protozoa, viral and UniVec kraken2 databases..

### 2.2. Building Prerequisite Files

For the root multi-FASTA file, we searched the archaea, bacterial, fungi, protozoa, and virus Kraken2 libraries for species from genera present in the MetaPhlAn3 markers database. Species not found in any Kraken2 libraries were downloaded as assemblies from NCBI (in assembly_summary_genbank.txt or assembly_summary_refseq.txt files available at ftp.ncbi.nlm.nih.gov/genomes/genbank/ (accessed on 5 September 2023)). Assemblies were chosen using parameters in the summary files: assemblies tagged as “excluded_from_refseq” were excluded from our files as well. If multiple assemblies were found in the RefSeq Category (“reference genome” > “representative genome” > then “NA”), then Assembly Level (“Complete Genome” > “Chromosome” > “Scaffold” > “Contig”) was used for selection. In cases of equivalence, the most recent assembly was used. Finally, if a genus still lacked a representative, we allowed for any available species in that genus. Here, “genus” refers to the NCBI taxID identified as “genus” (strict sense) or tagged as “no rank” or “clade” and placed between the species taxID and a taxID classified as “family”, “subfamily”, “tribe”, or “subtribe.” At the time of writing, 331 species lacked high-quality assemblies, and all but 19 species, from 18 genera, had at least one representative of their genus.

The correspondence file was built to link using the headers for all sequences in the multi-FASTA file with the taxonomic identifiers used by MetaPhlAn3. This was a record of species and genus classifications for each entry in the multi-FASTA.

### 2.3. Artificial Samples

For the CustomDB workflow performance evaluation, 100 artificial samples were generated using CAMISIM (v1.3) [20]. Of the 137 most abundant species in an unpublished study of African women, 100 were selected. Each had at least one assembly that was not already included in the prerequisite multi-FASTA file. The CAMISIM distribution parameters were set to a log-normal distribution with a mean of 1 (default setting). The standard variation was increased (set to 3). For diversity comparison, 50 artificial samples with low taxonomic and high taxonomic diversity were generated. Low diversity samples contained 10 *Faecalibacterium* and 8 *Bacteroides* species. The high taxonomic diversity samples contained 18 species from 18 different genera. The CAMISIM parameters were the same except for the standard variation of 2 (default settings). See Appendix A for details.

### 2.4. Phylogenetic Tree

Sequences for 10,202 16S rRNA gene sequences (of 13,610 in the multi-FASTA file) were downloaded from SILVA (v138.1) [21] and NCBI nucleotide databases (accessed on 5 September 2023). Sequences were aligned using MAFFT (Multiple Alignment using Fast Fourier Transform v7.505) [22] with the option-auto. RAxML (Randomized Axelerated Maximum Likelihood v8.2.12) [23] was used to construct the phylogenetic tree with the options: random seed (-p 1234), nucleotide substitution (-m GTRGAMMA), and 100 bootstraps (-N 100).

The 3408 remaining species, without a curated 16s sequence, were added as follows: The closest higher taxID of the one containing all target children was identified (i.e., family taxID of a genus). Where the taxID contained only one species, a new inner node was added midway between the leaf node and its parent node, and the missing species was added at the same distance as the original leaf from the newly added inner node. More complex cases required computing the Most Recent Common Ancestor (MRCA) and resulting rank purity and inclusiveness. The sum of purity and inclusiveness were maximized, with the constraint that ranked inclusiveness > 60%. The edge length of the new leaf node was the mean length of the already-present species in the genus to the maximized node. Illustrative examples can be found in Appendix A.

### 2.5. Diversity Calculations

The calculation α-diversity was performed using Shannon diversity and Faith’s phylogenetic distance [12,15]. β-diversity was computed using both Bray–Curtis dissimilarity and the weighted UniFrac methods [14,16]. α-diversities were calculated using reads classified using Kraken2 with the genus-level custom database. β-diversities were computed upon all classified reads and normalized by geometric means, as in DESeq2 (v1.42.0) [24]. All calculations were performed at the species level. The significance of the differences was evaluated using Mann–Whitney U statistical tests and *p*-values adjusted by False Discovery Rate correction with the Benjamini–Hochberg method.

### 2.6. Benchmark Comparisons

We used Diamond (v2.0.15) [25] as a modern mapper to compare our custom database. Diamond was paired with the curated UniProt (release 2023_01, April 2023) database [26] to perform accurate protein-based mapping of reads. A complement to this approach was MEGAN (Metagenome Analyzer, v6.21.14) [27], which was used to generate taxonomic identification based on Diamond reads. We used AMBER (Assessment of Metagenome BinnERs, v2.0.3) [28] to generate statistics and graphical outputs benchmarking mapping tools and the effects of database selection.

## 3. Results

### 3.1. A Project-Specific Database

We initiated the building of a specific project database to improve on methods that proved unable to confirm the deleterious effects reported for a high proportion of *Lactobacillus iners* in the vaginal microbiota during pregnancy versus that of *Lactobacillus crispatus* [29]. Specifically, we experienced difficulties in the correct identification of species in samples, including *Lactobacilli*, when using large inclusive databases, such as those for Kraken2 (Appendix A). Further, when using MetaPhlAn3 in order to achieve more precise species identification, we were unable to confirm the presence of *Klebsiella pneumoniae*, an important pathogen that had been isolated and identified using clinical microbiological methods on the same samples. We therefore sought to address the shortcomings of existing methods and to improve the power and accuracy of our analyses of the human microbiota. Our goal was to build a flexible pipeline capable of constructing a project-specific database that was adapted to any environment.

We called this workflow Moonbase, and the overall workflow is described below in Figure 1. Although MetaPhlAn3 failed to correctly identify some *Klebsiella* species in our analysis, we found that the overall identification, especially at the taxonomic level of genus, was on par or superior to other available tools [30]. As a starting point, all species from genera in MetaPhlAn3 were identified as a robust and inclusive set of genomes that could potentially be included in any custom database. As the foundation for building the custom database, we constructed a multi-FASTA from the list of 13,438 species and 3234 genera contained in MetaPHlAn3. The finalized multi-FASTA contained 752,507 sequences from 21,653 species = 13,610 bacteria + 665 archaea + 123 eukaryotes + 7255 viruses, with the additional species coming from those already included in the Kraken2 microbial database whose genus was in the MetaPhlAn list. In order to guard against changes in taxonomic nomenclature [31] and potential inconsistencies introduced through incremental updates at NCBI, we chose not to include the building of these files on users’ local machines. Instead, prerequisite files were prepared and stored in the Moonbase GitLab repository, namely the multi-FASTA file as well as a second file detailing the correspondence between the MetaPhlAn3 taxa species and the corresponding genome sequence. We were unable to find taxIDs and/or sequences for 18 of the 3234 genera, the majority of which were not relevant to human microbiota studies (Appendix A).

### 3.2. Running the Moonbase Pipeline

Moonbase was designed to build a project-specific database. The pipe accepted a range of data, including raw read files. Input files were data files in a format accepted by MetaPhlAn3, that is to say FASTA, FASTQ, bowtie2out, or SAM formats. Generally, these files correspond to the collection of samples for a given project. The workflow first runs MetaPhlAn3 on the samples to produce a file called merged_taxonomic_profiles.txt. This file contains, for each sample, the relative abundance of each species detected. Second, the user chooses which taxonomic level they wish to use for constructing the custom database. If species is used, then only those species explicitly identified by MetaPhlAn3 will be added to the txids_list_to_keep.txt file, which lists all the taxonomic identifications for species found across all input samples. If genus is selected, then all species from the parent genera identified from the input samples will be added to the list, even if the species in itself was not in the results of the MetaPhlAn 3.0 classification.

It then filters the multi-FASTA file to keep only sequences from taxIDs in the list. Finally, it builds the custom kraken2 database.

### 3.3. The Customized Database Improves Detection and Quantification

The performance of our project-specific database generated using the Moonbase pipeline was evaluated. We first generated a synthetic set of 100 samples using prevalent species found in the human microbiota using the CAMISIM software. We tested the taxonomic identification and quantification accuracy of MetaPhlAn3 as well as Kraken2 (generalized microbial database reference). These results were compared against Kraken2 run on the same samples using our custom database generated using either the ‘species’ or ‘genus’ level taxa parameter.

The results of the mapping comparison showed that read identification with Kraken2 using a generalized database performed relatively poorly when compared to ground truth. The performance of the generalized database was also poor compared to MetaPhlAn and with the Moonbase custom databases (Figure 2). Kraken2, when used with the generalized database, detected 32 of the 100 species present in the synthetic samples. Mapping with the custom species-level database generated with the Moonbase pipeline displayed significantly improved sensitivity and was successful in detecting 75 of 100. Sensitivity here was similar to the performance of MetaPhlAn3, where 74 of 100 species were detected. The similarity was expected and served as an effective control, as the species in the custom database, defined at the species level, were those explicitly identified by MetaPhlAn. Interestingly, when using the genus-level custom database, which included all members of the genus if a single species from that genus was identified by MetaPhlAn3, we detected 97 of 100 species. Our results showed that the most common species were identified in all tests. *Faecalibacterium prausnitzii*, with 500+ genomes in the NCBI database (https://www.ncbi.nlm.nih.gov/datasets/genome/?taxon=853 (accessed on 5 September 2023)) and nearly 1000 publications since 2002, was detected in all cases. This held true for other species that had been the subject of a relatively large number of studies and publications, including *Bacteroides thetaiotaomicron*, with 1000+ publications and 700+ genomes, and *Bacteroides fragilis*, with 6000+ publications and 800+ genomes. On the other hand, rarer taxa, including *Pseudoflavonifractor capillosus*, with only six references and none of the nine genomes sequences in the database complete, was only detected when using the genus-level custom database. *Faecalibacterium duncaniae* is a particularly instructive example in that it is a recently described species [32] in an otherwise well-established genus.

The correct identification of closely related species takes on added importance when considering the quantification of those species and when making comparisons across samples. We observed that, in many cases, the incorrect assignment of reads led to large errors in the calculation of abundances. This effect was most visible in the generalized microbial database using kraken2 (Figure 3). Here, we observed that, out of the ten *Faecalibacterium* species, only *Faecalibacterium prausnitzii* was detected with the generalized database, and this led to an overestimation of abundance of this species compared to the known abundance in the synthetic samples. Indeed, this bias in the calculation of abundance for *Faecalibacterium* was evident in MetaPhlAn and the species-level custom database. In general, as fewer species were correctly identified, those that were found tended to be assigned more reads, leading to both an overestimation of some species and an underrepresentation of overall diversity. Only in the case of the genus-level custom database did overall abundance and diversity closely match the known composition of samples.

### 3.4. Assessment of Correct Genome Assignment

Based on the results of species identification and quantification and the observed benefits of proper species-level genome identification, we benchmarked the ability of different mappers and different databases to correctly identify microbial species from complex mixed populations. We used AMBER to investigate the performance of different tools, including our own custom database coupled to Kraken2 mapping. AMBER is normally used to assess genome and taxonomic bins from metagenomic data. We harnessed its functionality in order to assess how well different methods and databases, including the Moonbase custom databases, correctly classified reads from 100 synthetic control samples. In this context, ‘bins’ were the 100 genomes used to construct the synthetic control samples with CAMISIM for which we knew the ground truth sequence as well as quantification. AMBER measures completeness, or the average amount that each genome is covered by mapped reads. Since control samples contained a range of species abundances, theoretical completeness is less than 100%, as some species did not have sufficient reads to completely represent the reference genome. Due to this fact and the fact that controls intentionally contained genomes not found in the prerequisite file, performance was best measured relative to other methods. AMBER also measured purity or the percentage of correctly mapped reads to each genome. The results for each genome bin averaged over all samples (Figure 4a) and the mean results of all bins for each method (Figure 4b) showed that Kraken2 mapping using our custom databases was significantly more performant than mapping with the larger non-specific microbial database. Mean completeness was 65.0% for the Moonbase species-level DB and 67.3% for the genus-level DB. Purity was 84.2% and 67.2%, respectively. This compared with a mean completeness of 31.7% and a purity of 41.0% for the general Kraken2 DB.

Interestingly, we found that our custom database performed as well as protein mapping with Diamond, using the curated UniProt database, in terms of purity, and was able to recruit a higher number of reads that resulted in higher completeness (Figure 4c,d). Diamond-UniProt and MEGAN produced 79.3% and 52.9% purity scores, coupled to 47.3% and 35.1% completeness scores, respectively. Although protein mapping experiences a built-in penalty due to the fact that not all genomic sequences are codes of proteins, the targeted Moonbase DBs still appeared to outperform protein mapping with high-quality but generalized databases.

### 3.5. Whole-Genome Metagenomic Diversity with Phylogeny

As with the detection of microorganisms and their abundance, diversity is a key parameter in the analysis of microbial communities. The use of phylogenetic information to calculate this diversity for microbial samples was introduced in the form of UniFrac analysis. UniFrac uses the distances in a phylogenetic tree generated using 16s rRNA genes to estimate overall genetic diversity. While tools such as Unifrac, for integrating phylogenetic information, are easily applicable to 16s molecules, the same cannot be said for the analysis of whole-genome shotgun metagenomic sequencing. However, we found that, with a defined set of genomes used to build our custom database, we could also construct a phylogenetic tree using 16s rRNA genes from these same representative sequences.

We were able to identify and download a combined 10,202 16S rRNA gene sequence, from the SILVA and NCBI nucleotide databases, out of a total of 13,610 bacteria species. MAFFT was used to align the sequences and RAxML was subsequently employed to generate the phylogenetic tree. Construction of the 16s rRNA gene tree required approximately seven days of computational time. The placement of the other 3408 species is described in the Methods section. Briefly, we identified the Most Recent Common Ancestor (MRCA) and determined the position of the new leaf node by maximizing purity and inclusiveness of the target species, exemplified in Appendix A. The final tree is available in the project repository and can be used to calculate diversity.

### 3.6. Inclusion of Phylogenetic Diversity Increases Statistical Power

In order to determine how the inclusion of phylogenetic data could contribute to increased statistical power, we examined the α-diversities and β-diversities of two groups of CAMISIM-generated synthetic samples (Appendix A). Samples with 18 distinct species differed in the number of parental genera; LD-18 (low-diversity, *n* = 50) samples were comprised of two genera while HD-18 (high-diversity, *n* = 50) samples were comprised of 18 unique genera. These compositions were used to approximate microbial communities, such as the vaginal microbiota, where one species of the Lactobacillus genus might be mixed or replaced by another species from the same genus, or by more distantly related species that are known to increase the risks of dysbiosis.

We computed α-diversities, or the measurement of diversity and evenness within samples, using the Shannon method (Figure 5a). We compared this diversity calculation with Faith’s phylogenetic diversity (PD) method, which incorporates phylogenetic information (Figure 5b). For Faith’s PD, the distances of the branches were provided by the phylogenetic tree described above. Shannon diversity calculations incorporate the number and evenness of individual taxa but are blind to the genetic composition of these taxa. A Mann–Whitney U test, with Benjamini–Hochberg FDR correction, comparing Shannon α-diversities between LD-18 and HD-18 samples identified no significant differences between the two groups. This was not surprising since the number of species and abundance distributions parameters were equal for both sets of samples. However, when Faith’s phylogenetic diversity metric was used, a calculation that does account for phylogenetic distances in the calculation of diversity, a significant difference was identified (Mann–Whitney U with FDR *p*-value = 1.84–18).

We performed a similar comparison of β-diversities, which measure the differences in community composition between samples. Pairwise comparisons were performed in order to calculate a distance matrix for all samples. Analysis of metagenomic samples commonly uses Bray–Curtis dissimilarity as a means of calculating distance. This measurement handles sparce datasets with many zeros while returning a bounded range of values between zero and one. Bray–Curtis dissimilarity integrates counts but not phylogenetic information. Figure 5c displays a plot of Bray–Curtis dissimilarity β-diversities for LD-18 and HD-18 samples. No statistically significant differences were observed between the two sets of samples. An alternative method of calculating the distance matrix is by using weighted UniFrac. UniFrac, like Bray–Curtis, returns a value between zero and one, and the weighted version is sensitive to changes in abundance. In addition, UniFrac can account for evolutionary relationships. Figure 5d shows a plot of β-diversities calculated with weighted UniFrac. Using the same Mann–Whitney U test, we find a corrected *p*-value to be highly significant when comparing LD-18 and HD-18 samples (*p*-value = 4.78–153). Taken together, these results demonstrate that the addition of phylogenetic information to calculations of diversity can increase the sensitivity and statistical power of analyses.

## 4. Discussion

### 4.1. An Improvement over a Large General Database for Metagenomics

Recent work identifying *Klebsiella pneumoniae*/*Kocuria rosea* as a possible additional community state type (CST) reinforced our previous experience in quantification potential pathogens in the vaginal microbiota [4,33]. The results in this work demonstrate further potential for biases in quantification for microbiota samples. For the *Faecalibacterium genus* (Figure 2 and Figure 3), the most intensively studied and sequenced species, *Faecalibacterium prausnitzii*, was readily detected. However, recently isolated species (e.g., *Faecalibacterium* sp. *I3389* and *Faecalibacterium* sp. *I4384*), which could be important for novel metabolic functions, were only detected using our custom database [32]. *Pseudoflavonifractor capillosus*, a species associated with the use of proton pump inhibitors, was recently reclassified, and is another apt example where a custom database built at the genus level is the only method able to provide accurate quantification [34,35]. Our results confirmed that building the custom database at the genus level offered the best precision and sensitivity for most use cases.

Building a custom database from a fixed set of genomes stored in the prerequisite file has allowed for the inclusion of significant additional functionality. The assessment of diversity in microbial populations is often critical to understanding the function and stability in a given environment [36]. We have shown here that incorporating taxonomic distances in α- and β-diversities contributes significantly to the power of the analyses being performed. Here, we have described the use of the 16S rRNA genes to build a tree for the calculation and use of taxonomic information. This taxonomic information is otherwise difficult to assess and is routinely ignored in standard whole-genome metagenomic analysis. A further example of added functionality is the improved accuracy of quantification of the taxa present in sequenced samples through appropriate normalization of data. Bracken normalization, which is used to correct read quantification based on Kraken mapping, is performed based on a correction file built from a custom database in comparison to its component genomes. This type of correction is especially pertinent in human microbiota studies, where correct quantification of closely related species can mean the difference between pathogen and commensal [37,38]. The fixed set of potential species in the Moonbase files allows all constructed custom databases to have the necessary genome files in order to perform Bracken normalization based on database composition.

### 4.2. Perspectives and Improvements

The decision to build the Moonbase custom database from a predetermined set of species presents challenges in keeping data updated to reflect the latest results in sequencing and taxonomic assignments. The advantages of a custom database rely on having access to the most up-to-date sequencing data and taxonomic annotations. This can only be achieved through regular updates. The custom databases generated by Moonbase are ultimately dependent on the composition of the individual samples, as determined by the integrated MetaPhlAn component. As such, it is important that future updates to the MetaPhlAn analysis pipeline be reflected in the Moonbase pipeline. In constructing the database, we found one of the most challenging tasks to be resolving the often-conflicting taxonomic identifications across MetaPhlAn3, Kraken2, NCBI, and SILVA databases. Our ultimate solution was to include a dedicated file to make the necessary correspondence for all possible genomes that could be included in the database and 16s tree. While preparing an update to the recently published MetaPhlAn4 database, we will explore the integration of the species and alignments from the All-Species Living Tree Project (LTP) [39]. LTP includes both Bacterial and Archaeal species, and we plan to include Archaea in a future version of the tool. Although the number of archaea in the human microbiota is small, methanogens have been isolated from several sites and have been shown to play an important role in metabolism in the intestinal microbiota [40,41]. It is our hope that standardization and simplification of the update process will allow us to generate annual updates to the Moonbase tool. The Moonbase repository on GitLab will serve as a means to provide future updates and introduce an added function to our pipeline.

## 5. Conclusions

Our development and in-depth testing of the Moonbase pipeline underscores the importance of using properly tailored databases appropriate for the types of metagenomic analyses being performed and the environments being studied. This observation is particularly critical for capturing less studied but potentially significant microbial species in complex communities. The Moonbase custom database not only enhances the precision and sensitivity of species identification but also allows for nuanced analyses like taxonomic distance-based diversity assessments. As sequence databases push into tera- and peta-base scales, project-specific databases, such as Moonbase, capable of harnessing and focusing available data for researchers, will become an ever more important resource.

## Figures and Tables

**Figure 1 microorganisms-12-00135-f001:**
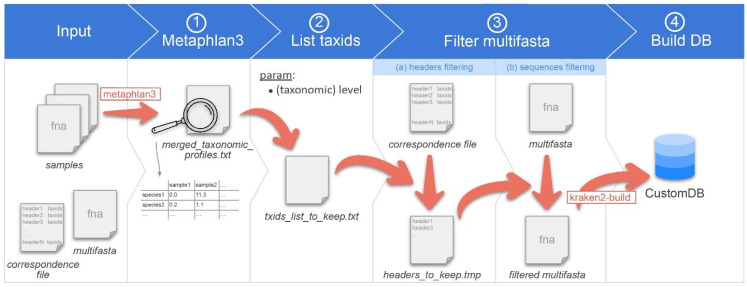
Overview of the CustomDB creation snakemake workflow. Overview of the CustomDB pipeline. Input is provided in the collection of samples in a format accepted by metaphlan3 (FASTQ, FASTA, bowtieout or sam). A multi-FASTA file and taxonomy correspondence file are also required. (1) MetaPhlAn 3.0 is run on samples, and the results merged into the merged_taxonomic_profiles.txt file. (2) The list of all taxIDs present in the samples is made at the species or genus level, depending on the value of the corresponding user-defined parameter. (3) The multi-FASTA file is filtered, retaining only sequences from taxIDs listed in the output file of step #2. (4) The finalized database was built using kraken2-build.

**Figure 2 microorganisms-12-00135-f002:**
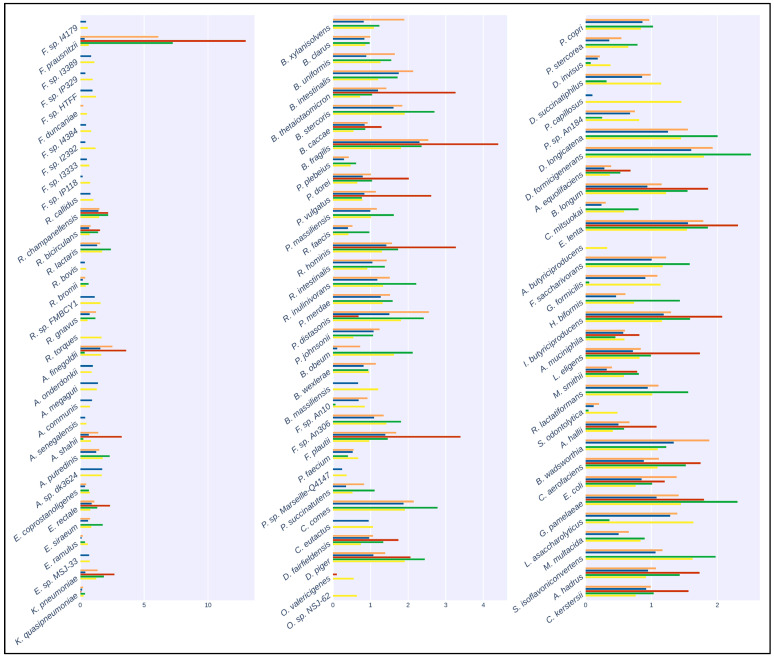
Taxonomic classification of the artificial samples generated by CAMISIM. Bar plots show the relative abundance of the 100 species present in the artificial samples. Colors correspond to: Orange: Kraken2 with CustomDB-species, Blue: Kraken2 with CustomDB-genus, Red: Kraken2 with generalized microbial database, Green: MetaPhlAn3, Yellow: Ground Truth.

**Figure 3 microorganisms-12-00135-f003:**
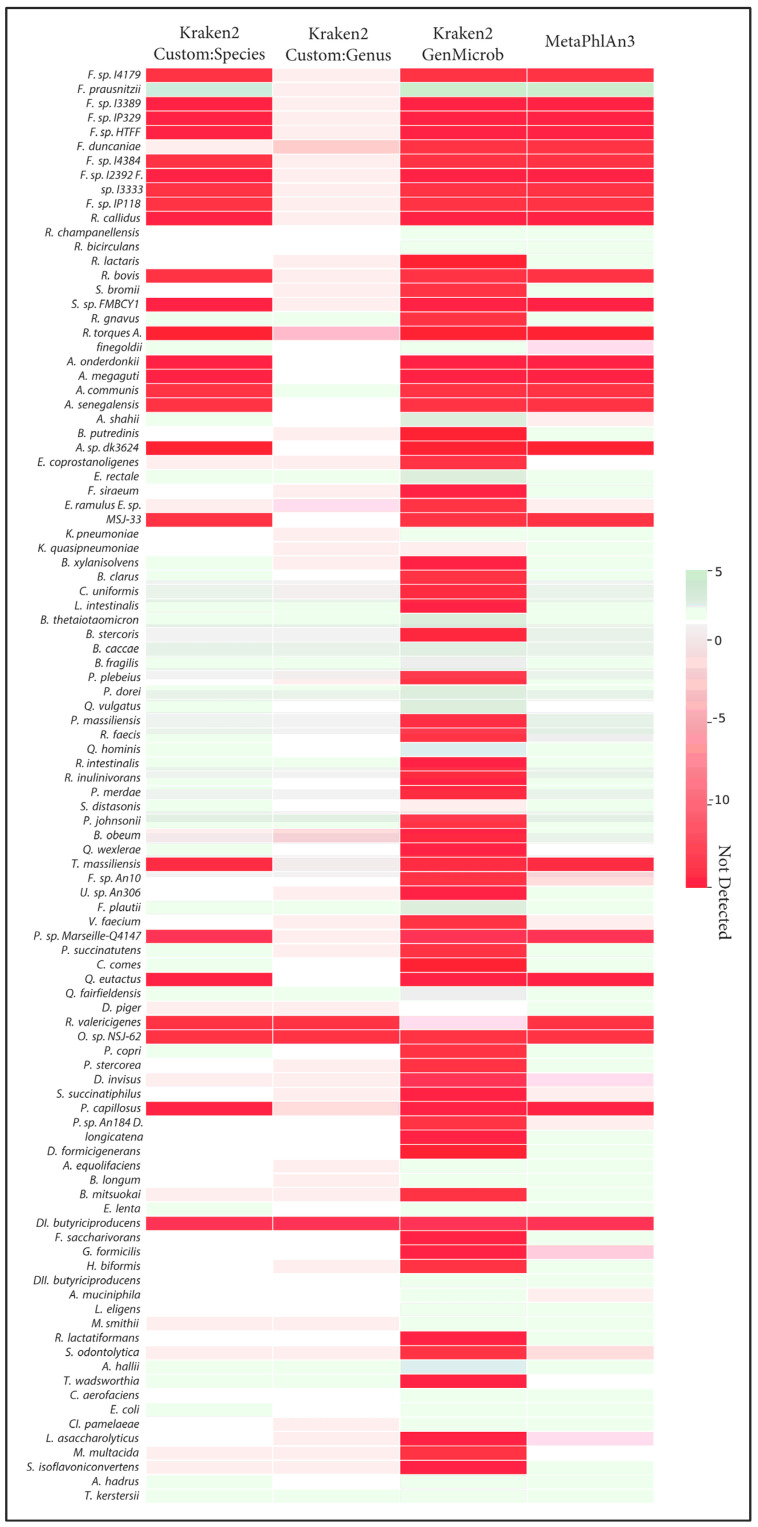
Heatmap of the Log2 differences in relative abundance of detection of the 100 species from Figure 1 by different methods; Kraken2 (CustomDB-species), Kraken2 (CustomDB-genus), Kraken2 (generalized microbial DB), and MetaphlAn3 are compared against ground truth. Red indicates errors of underestimation or absence of taxa, and green denotes errors of overestimation.

**Figure 4 microorganisms-12-00135-f004:**
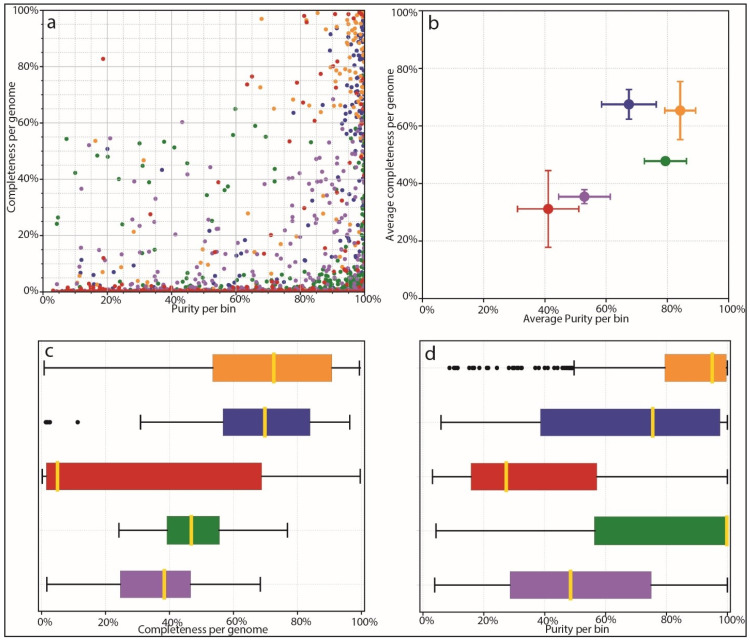
AMBER assessment of binning. Binning methods are colored as follows: Orange: Kraken2 mapping using CustomDB-species, Blue: Kraken2 mapping using CustomDB-genus, Red: Kraken2 mapping using the generalized microbial database, Green: Diamond mapping (UniProt), Purple: MEGAN taxonomy. (**a**) Dot plot of binning results from each mapper showing completeness and purity. Ground-truth bins located in the top right of the plot are all 100% complete with 100% purity. (**b**) Average completeness and purity positions for groups of bins from ‘A’, giving equal weight to all bins. (**c**) Bar plot comparing the completeness of bins for each method. (**d**) Bar plot comparing purity of bins for each method.

**Figure 5 microorganisms-12-00135-f005:**
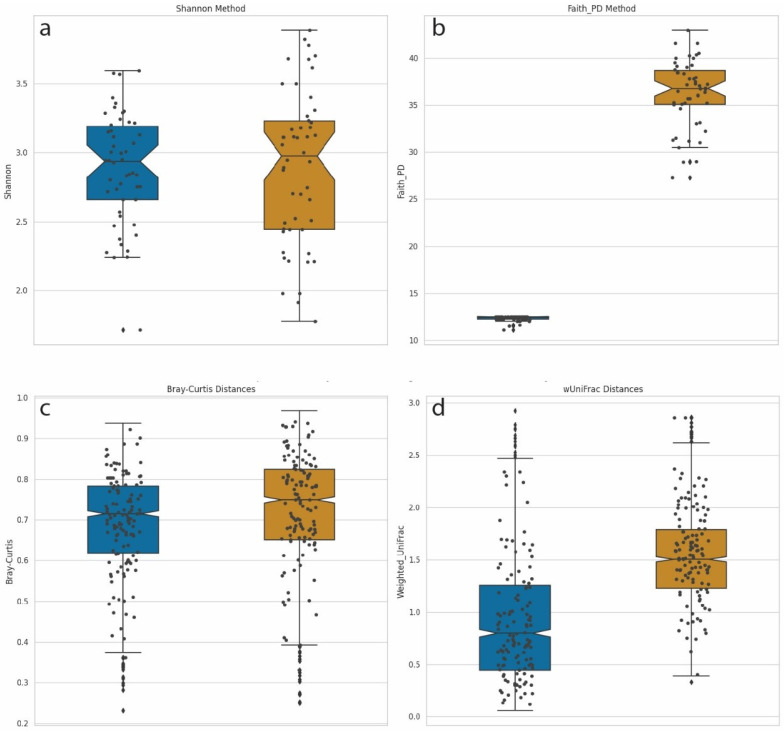
Box-and-whisker plot comparisons of diversity calculation methods. Left/Blue plots = Synthetic samples built from 18 species from two genera: Dots denote individual samples. LD-18, Right/Gold = Synthetic samples built from 18 species from 18 genera: HD-18 (**a**) Shannon α-diversity for 100 samples from both LD-18 and HD-18. (**b**) Faith’s PD α-diversity for 100 samples from both LD-18 and HD-18. (**c**) Bray–Curtis dissimilarity calculation of β-diversity for LD-18 and HD-18 samples. (**d**) Weighted UniFrac distances calculation of β-diversity for LD-18 and HD-18 samples.

## Data Availability

The code and files used for installing and running Moonbase are available on our Gitlab (https://gitlab.pasteur.fr/metagenomics/moonbase). The synthetic data set used for validation are available upon request.

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
