# Peer review of "Targeted Metagenomic Databases Provide Improved Analysis of Microbiota Samples"

_microorganisms, 2024, doi:10.3390/microorganisms12010135_

Round 1
Reviewer 1 Report
Comments and Suggestions for Authors
I have revised the text for you. Here is the improved version:
The paper is well written and proposes a solution to create databases using the Moonbase pipeline, which is supposed to improve the precision and quality of identifications in metagenomic analyses. The improvements obtained using the Moonbase pipeline are significant, as they double the performance of existing solutions in some scenarios. The authors say: “Mean completeness was 65.0% for the Moonbase species-level DB and 67.3% for the genus-level DB. Purity was 84.2% and 67.2%, respectively. This compares with a mean completeness of 31.7% and purity of 41.0% for the general Kraken2 DB.” This is certainly interesting and a good achievement, but they do not reach near perfection. I have not found a clear explanation for why synthetic samples made of known genomes are not perfectly identified (or almost) when using their database. Can they explain the reasons behind that and how they think that this could be achieved?
Also, they correctly state that, in order for the database to be useful, it must be updated and enlarged to encompass all the known diversity, on a regular basis. Do they have clear plans on how to make this sustainable in the long term?
I have not checked the English much but I found a few mistakes, such as “relay” instead of “rely” for example.
Author Response
Comment:
The paper is well written and proposes a solution to create databases using the Moonbase pipeline, which is supposed to improve the precision and quality of identifications in metagenomic analyses. The improvements obtained using the Moonbase pipeline are significant, as they double the performance of existing solutions in some scenarios. The authors say: “Mean completeness was 65.0% for the Moonbase species-level DB and 67.3% for the genus-level DB. Purity was 84.2% and 67.2%, respectively. This compares with a mean completeness of 31.7% and purity of 41.0% for the general Kraken2 DB.” This is certainly interesting and a good achievement, but they do not reach near perfection. I have not found a clear explanation for why synthetic samples made of known genomes are not perfectly identified (or almost) when using their database. Can they explain the reasons behind that and how they think that this could be achieved?
Response:
We appreciate the positive comments concerning our pipeline and the improvement that can be obtained in accurate identification of reads. We agree that the reason for <100% purity and completeness performance should be further clarified in the manuscript.
Completeness, or the complete coverage of a given reference genome, depends on the abundance of reads mapped of the full length to that reference. There are three explanations:
- Our synthetic samples contained species mixed at high and low abundances, across three logs values (as stated in methods). Low-abundance species do not have a sufficient number of reads to achieve full completeness since they will not cover the full genome of reference sequences. 100% completeness is not possible.
- Below 100% performance is also due to the complexity of metagenomic samples, and species which may be closely related. This effects the generalized Kraken2 DB (previously called ‘fatfree’) to a greater extent. This second issue is also the primary reason purity does not achieve a perfect 100%.
- Finally, some species might not be represented our prerequisite multi-FASTA file (stated in methods).
Purity and completes are best, therefore, compared relative to other method of detections.
The text has been modified as follows:
The original phrase: “In our case, maximum theoretical completeness is less than 100% since the control samples were generated varying taxa abundances.”
Modified wording: “Since control samples contained a range of species abundances, theoretical completeness is less than 100% as some species do not have sufficient reads to completely represent the reference genome. Due to this fact, and that controls intentionally contained species not found in the prerequisite file, performance is best measured relative to other methods.”
Comment:
Also, they correctly state that, in order for the database to be useful, it must be updated and enlarged to encompass all the known diversity, on a regular basis. Do they have clear plans on how to make this sustainable in the long term?
Response:
We thank the reviewer for pointing out this issue. The use of the GitLab repository was a means by while the code and accompanying file could be updated to the community. It will also allow for “pull requests” from members outside of the team to address bugs and feature requests.
We have added the following text at the end of section 4.2: “The Moonbase repository on GitLab will serve as a means to provide future updates and introduce added function to our pipeline.”
Comment:
I have not checked the English much but I found a few mistakes, such as “relay” instead of “rely” for example.
Response:
We have corrected the grammatical error (section 4.2) and also reviewed to text for other such errors.
Reviewer 2 Report
Comments and Suggestions for Authors
Dear authors,
thanks for these works.
It wasn't immediately clear to me what are the novel aspects of these works and what are derivatives/ assembly of other tools. E.g., your pipeline seems heavily dependent on Metaphlan (and then kraken).
Also, would your approach imply that one has to build custom databases for one's projects, perhaps using your pipeline? It did not come across so clearly on the application of these works.
Introduction is very non-specific and does not not set up the ensuing works presented here. On the other hand, I found that lines 161-176 do not belong to Results but rather are very well suited for Introduction section. Similarly, section 4.1 lines 384 - 404 would also have found better placement in Introduction section to explain the background and rationale behind these works.
What is meant by microbial fatfree database (line 225). The fatfree connotation is not clear.
Figure 2 is not legible at all in current presentation when printed on a usual A4 size
References 42-49 need be removed from text since nothing cited here in.
Author Response
Reviewer #2
Comment:
It wasn't immediately clear to me what are the novel aspects of these works and what are derivatives/ assembly of other tools. E.g., your pipeline seems heavily dependent on Metaphlan (and then kraken).
Also, would your approach imply that one has to build custom databases for one's projects, perhaps using your pipeline? It did not come across so clearly on the application of these works.
Response:
Thank you for identifying this lack of precision. We have updated the first lines of the abstract (lines 7-10) to make the point clear that our pipeline uses both MetaPhlAn and Kraken2 and that its purpose is to improve Kraken2 mapping by generating tailored databases:
“We report on Moonbase, an innovative pipeline the builds upon the established tools of MetaPhlAn and Kraken2, enhancing their capabilities for more precise taxonomic detection and quantification in diverse microbial communities. Moonbase enhances the performance of Kraken2 mapping by providing an efficient method for constructing project-specific databases.
Comment:
Introduction is very non-specific and does not not set up the ensuing works presented here. On the other hand, I found that lines 161-176 do not belong to Results but rather are very well suited for Introduction section. Similarly, section 4.1 lines 384 - 404 would also have found better placement in Introduction section to explain the background and rationale behind these works.
Response:
We thank the reviewer for their specific comments on improving the introduction. The first lines of the results (previously 161-176 ) section have been move to the third paragraph in the introduction. The majority of the beginning of the discussion (lines 384 – 400) now constitute the forth paragraph in the introduction. The revised introduction (lines 57-71) now provides a better balance of broad overview and more specific information about the work in the manuscript. Results and discussion are likewise more focused.
Comment:
What is meant by microbial fatfree database (line 225). The fatfree connotation is not clear.
Response:
Thank you for pointing this out. ‘Fatfree’ was a general way of describing a general microbial database that includes archaea, bacteria, fungi, protozoa and viral sequences. This language is not informative and so the term ‘fatfree database’ has been replaced with ‘generalized microbial database’ for better clarity. Section 2.1 of material and methods now also states:
(Line 108-110) “The ‘generalized microbial database’ used for comparing Kraken2 performance was generated using the kraken2 --download-library command for archaea, bacteria, fungi, protozoa and viral database types.”
Comment:
Figure 2 is not legible at all in current presentation when printed on a usual A4 size
Response:
We have made two adjustments to correct this.
- Because of the size of the figure, we split it into two separate figures so each can be slightly larger and more readable. Figure 2A+B is now Figure 2 and Figure 3, with the remainder of figures renumbered and text modified accordingly. Figure legends were modified accordingly.
- In both revised figures, we abbreviated the genus name to allow for a slightly larger font to be used for species. The resolution of figures was also increased to 300ppi (previously 150 ppi).
Comment:
References 42-49 need be removed from text since nothing cited here in.
Response:
Indeed, they were left over from copying into the submission format. These extraneous references have been deleted.
Round 2
Reviewer 2 Report
Comments and Suggestions for Authors
Thanks for your response to my remarks